# *Xanthomonas* Phage PBR31: Classifying the Unclassifiable

**DOI:** 10.3390/v16030406

**Published:** 2024-03-06

**Authors:** Rashit I. Tarakanov, Peter V. Evseev, Ha T. N. Vo, Konstantin S. Troshin, Daria I. Gutnik, Aleksandr N. Ignatov, Stepan V. Toshchakov, Konstantin A. Miroshnikov, Ibrahim H. Jafarov, Fevzi S.-U. Dzhalilov

**Affiliations:** 1Department of Plant Protection, Russian State Agrarian University-Moscow Timiryazev Agricultural Academy, Timiryazevskaya Str. 49, 127434 Moscow, Russia; r.tarakanov@rgau-msha.ru (R.I.T.); konstantinetr@gmail.com (K.S.T.);; 2Shemyakin-Ovchinnikov Institute of Bioorganic Chemistry, Russian Academy of Sciences, Miklukho-Maklaya Str., 16/10, 117997 Moscow, Russia; 3Laboratory of Molecular Microbiology, Pirogov Russian National Research Medical University, Ostrovityanova 1, 117997 Moscow, Russia; 4Faculty of Agronomy, Nong Lam University, Quarter 6, Thu Duc District, Ho Chi Minh City 721400, Vietnam; 5Limnological Institute, Siberian Branch of the Russian Academy of Sciences, 664033 Irkutsk, Russia; daria_gutnik@mail.ru; 6Agrobiotechnology Department, Agrarian and Technological Institute, RUDN University, Miklukho-Maklaya Str. 6, 117198 Moscow, Russia; ignatov_an@pfur.ru; 7Center for Genome Research, National Research Center “Kurchatov Institute”, Kurchatov Sq., 1, 123098 Moscow, Russia; 8Azerbaijan Scientific Research Institute for Plant Protection and Industrial Crops, AZ 4200 Ganja, Azerbaijan

**Keywords:** *Xanthomonas* phage PBR31, bacteriophage taxonomy, temperate phages, *Xanthomonas campestris* pv. *campestris*, phage control

## Abstract

The ability of bacteriophages to destroy bacteria has made them the subject of extensive research. Interest in bacteriophages has recently increased due to the spread of drug-resistant bacteria, although genomic research has not kept pace with the growth of genomic data. Genomic analysis and, especially, the taxonomic description of bacteriophages are often difficult due to the peculiarities of the evolution of bacteriophages, which often includes the horizontal transfer of genes and genomic modules. The latter is particularly pronounced for temperate bacteriophages, which are capable of integration into the bacterial chromosome. *Xanthomonas* phage PBR31 is a temperate bacteriophage, which has been neither described nor classified previously, that infects the plant pathogen *Xanthomonas campestris* pv. *campestris*. Genomic analysis, including phylogenetic studies, indicated the separation of phage PBR31 from known classified bacteriophages, as well as its distant relationship with other temperate bacteriophages, including the *Lederbervirus* group. Bioinformatic analysis of proteins revealed distinctive features of PBR31, including the presence of a protein similar to the small subunit of D-family DNA polymerase and advanced lysis machinery. Taxonomic analysis showed the possibility of assigning phage PBR31 to a new taxon, although the complete taxonomic description of *Xanthomonas* phage PBR31 and other related bacteriophages is complicated by the complex evolutionary history of the formation of its genome. The general biological features of the PBR31 phage were analysed for the first time. Due to its presumably temperate lifestyle, there is doubt as to whether the PBR31 phage is appropriate for phage control purposes. Bioinformatics analysis, however, revealed the presence of cell wall-degrading enzymes that can be utilised for the treatment of bacterial infections.

## 1. Introduction

Bacteriophages (also known as “phages”) are viruses that specifically infect bacteria. They are ubiquitous and can be found in water, soils and many living organisms [1]. Some estimates indicate that phages can cause approximately 10^18^ successful infections in a single second [2]. The total number can be approximated to be 10^3^¹ virions, which outnumbers bacterial cells by a factor of 10 to 100 [3]. Their combined mass is estimated to be about a trillion tons [4].

The ability of bacteriophages to destroy pathogenic bacteria attracted the attention of researchers in the first half of the 20th century, and in recent decades, interest in their use as therapeutics, primarily due to the rise of antibiotic resistance in bacteria, has been revived. Phage therapy/phage control offers important benefits, including a high specificity for bacterial targets and the potential for minimal harm to the cells and tissues of macroorganisms [5].

Growing interest in bacteriophage research and application necessitates improvement in phage characterisation methods, including taxonomic assignment. Historically, phages have been classified on the basis of particle morphology, but at the time of the first classification schemes describing bacteriophages, PCR, sequencing and the many molecular techniques we now know were not available [6,7]. A new classification scheme for tailed bacteriophages, based on genomic data, was adopted by the International Committee on Taxonomy of Viruses (ICTV) in 2021, eliminating families defined by morphological characteristics [8]. All tailed phages were assigned to the class *Caudoviricetes*, which belonged, in turn, to the type *Uroviricota* of the kingdom *Heunggongvirae* of the realm *Duplodnaviria*. In 2022–2023, the new system was further refined, with the emergence of new high-ranking taxa at the level of subfamilies and higher, as well as new genera [9,10]. However, most of the more than 20,000 bacteriophages known to science, whose genomes are presented in the NCBI GenBank database, have not yet been classified, which is partly due to difficulties in substantiating the taxonomic descriptions of phages. This is also due to the fact that taxonomy has not kept pace with the growth of genomic data. Another reason is related to the characteristic nature of viral evolution, accompanied by the rapid accumulation of mutations and the chimeric origin of some viral genomes [11,12,13]. The latter feature is especially pronounced for temperate phages, which are capable of integrating their genome into the bacterial chromosome [14,15].

Bacteriophages infecting phytopathogenic *Xanthomonas* sp. are particularly significant, since they can be used for the biocontrol of these bacteria. The ability of bacteriophages to lyse pathogen cells, and challenges in controlling bacterial diseases in plants, have aroused interest in the application of phage therapy for black rot management [16]. Black rot in brassicas, caused by a Gram-negative *Xanthomonas campestris* pv. *campestris* (*Xcc*), is the most important disease affecting brassica crops worldwide, but the pathogen can cause root rot of winter oilseeds, as well [17]. Recommendations for phage therapy suggest choosing strictly lytic phages, primarily to avoid the potential transfer of antibiotic resistance or virulence genes by temperate phages through transduction, but synthetic biology can be used to enhance the safety and efficacy of temperate phages [18]. Moreover, phage enzymes that degrade the host phage cell wall could also be utilised for the treatment of bacterial infections [19].

*Xanthomonas* phage PBR31 is a presumably temperate tailed phage infecting *Xcc* which was first isolated in Moldova in 2014. Interestingly, the identical phage was also found in the Moscow region, later. The occurrence of *Xanthomonas* phage PBR31 in geographically distant areas may indicate the high level of adaptation of this phage to parasitism on seed-born *Xcc*, causing particular interest in this phage. The purpose of the present study was to introduce the *Xanthomonas* phage PBR31 and to describe its biological and genomic features. First, the biological properties of the phage were described. Next, the phage genes and proteins were predicted and analysed. Then, based on genomic features, the taxonomic analysis was carried out. Finally, the implications of the analyses were discussed. The aim of this study was to provide a complete biological, genomic and taxonomic description of phage PRB31 and to identify its features that may be useful for both theoretical and practical use.

## 2. Materials and Methods

### 2.1. Bacteriophage Isolation and Purification

Bacteriophage PBR31, specific against *Xcc*, was isolated from a mixture of soil and stumps after harvesting cabbage using the Ram3-1 strain, according to [20], with modifications. A soil sample with stumps was homogenised, suspended in King’s B broth and shaken for 30 min at 26 °C on an orbital shaker. Soil particles were removed by centrifugation at 4000× *g* for 30 min and the supernatant was sterilised with a 0.22 μm pore size filter (Labfil, ALWSCI Technologies, Hangzhou, China). A 1 mL solution was added to a 50 mL tube containing 25 mL King’s B broth and 100 µL of *Xcc* Ram3-1 overnight culture, and left in an orbital shaker. After 12 h, bacterial cells were removed by centrifugation at 4000× *g* for 30 min, the supernatant was sterilised with a filter (pore size 0.22 μm) and 5 μL was added to the top agar with strain Ram3-1. After incubation at 26 °C overnight, bacteriophage spots were identified. An agar plug containing one spot was transferred, with a loop, into 900 μL of SM buffer (50 mM Tris/HCl, pH 7.5, 100 mM NaCl, 10 mM MgSO_4_ and gelatine to 0.01%) [21] and homogenised by vortex; then, CHCl_3_ was added and three purification cycles were carried out using the serial dilution method. The resulting isolate was purified by centrifugation at 8000× *g* for 20 min, followed by filtration of the supernatants through membrane filters with a pore size of 0.22 μm, and then the addition of DNase I (0.5 mg/mL, 1 h; Evrogen, Moscow, Russia). The resulting filtrate was concentrated by ultracentrifugation at 100,000× *g* at 4° C for 2 h using a Beckman SW28 rotor (Beckman Coulter, Brea, CA, USA). The final purification of the phage suspension was carried out by ultracentrifugation in a step density gradient of CsCl (0.5–1.7 g/mL) at 22,000× *g* for 2 h. The resulting opalescent band was dialysed against SM buffer and the phage suspension was stored at 4 °C.

### 2.2. Determination of Phage Host Range

The lytic activity and host range of the phage were determined by applying a phage sample to a lawn containing *Xcc* strains. An amount of 200 μL of the appropriate strain was mixed with 4 mL of King’s B top agar (0.7%), which had been prewarmed to 48 °C, and poured onto a King’s B agar plate [22]. After drying in a laminar flow hood for 10 min, 10 μL of purified phage suspension was dropped onto the medium, and after 24 h of cultivation at 28 °C, the formation of spots was observed. A transparent zone at the site of application indicated the lytic action of the phage against this strain. For additional verification, the suspension was titrated in SM buffer and dripped in lines onto the top agar with the same strain; if the reaction was positive, the formation of single plaques was observed.

### 2.3. Phage Adsorption and One-Step Growth Experiments

A one-stage growth curve was plotted according to [23], with modifications. Overnight, *Xcc* Ram3-1 suspension (9 mL) was adjusted to a 10^6^ colony-forming unit (CFU) per mL, mixed with 1 mL of PBR31 phage solution at multiplicity of infection (MOI) 0.01 and incubated at 26 °C for 60 min. The mixture was centrifuged at 12,500× *g* for 2 min to remove unadsorbed phage virions. The pellet was suspended in King’s B broth and incubated at 26 °C and 200 rpm agitation. Every 20–40 min, 100 μL of the supernatant was taken, purified with chloroform and diluted tenfold in SM buffer, and then the titre was determined on King’s B top agar with Ram3-1 after 24 h of cultivation in a thermostat at 26 °C. The burst size of the phage was determined as the ratio of the average number of free phage particles after the release phase (plateau average (PFU/mL)) to the corresponding number of phage particles (PFU/mL) added to the exponentially growing bacterial cells. The experiment was repeated three times and a one-step growth curve was plotted.

The phage adsorption curve was plotted as described in [24]. Ram3-1 cells grown in King’s B broth overnight were mixed in a sterile tube with fresh King’s B broth and phage PBR31 at an MOI of 0.01. The mixture was placed on a shaker–incubator and cultured at 26 °C and 200 rpm. After 3–100 min, 50 μL of the suspension was taken and diluted 100 times in 4.45 mL of King’s B broth, and 0.5 mL chloroform was added. The resulting mixtures were mixed by vortex, kept for 40 min at room temperature and titrated to determine the number of unattached phages. The adsorption curve was plotted according to the ratio of unadsorbed phages at different time intervals to the initial number of phages. The experiment was repeated three times.

### 2.4. Phage Stability under Different Conditions

The stability of the phage under different conditions was studied according to [25], with some modifications. A phage suspension with a titre of 10^6^ PFU/mL was prepared in SM buffer. To determine phage stability at different temperatures, phage suspensions were incubated at 4, 10, 20, 30, 40, 50, 60, 70, 80, 90 and 100 °C for 1 h in a Thermomixer F 2.0 (Eppendorf, Hamburg, Germany). To assess the resistance of the phage to ultraviolet radiation, the samples were illuminated with a PL-S9W/12/2p UV lamp (Philips, Amsterdam, The Netherlands) (280–315 nm) for 90 min, and 50 μL samples were placed into separate tubes every 10 min. To assess the stability of the phage to different acidities, a series of buffer solutions (20 mM Tris-HCl/20 mM Na-citrate/20 mM Na-phosphate), adjusted with NaOH to a pH in the range of 3–12, was added to the samples, up to 10^6^ PFU/mL phage and incubated at 25 °C for 1 h. To assess the sensitivity of the phage to chloroform, solutions of the phage and chloroform were mixed to a concentration of 5%, 25%, 50% and 75% of the volume. Then, the tubes were shaken vigorously and incubated at 26 °C for 30 min, according to [26].

The solutions obtained in the experiments were centrifuged at 8000 rpm for 15 min and the phage layer was collected. The solutions were titrated on King’s B top agar with Ram3-1 and after 24 h of cultivation the titre in each sample was determined. Each experiment included three parallel repetitions.

### 2.5. Calculation of MOI

Determination of the optimal MOI was carried out according to [27], with modifications. To do this, sterile King’s B broth, 100 μL of an overnight culture of *Xcc* Ram3-1 and phage were added to sterile tubes at a multiplicity of infection of 0.001, 0.01, 0.1, 1, 10, 100 and 1000. The solutions were incubated at 26 °C, with shaking (100 rpm) on an orbital shaker for 24 h, and centrifuged at 11,000× *g* for 10 min. The resulting solutions were titrated in a similar way to that described in Section 2.6. The experiment was repeated three times. The multiplicity of infection with the highest titre was the optimal multiplicity of infection (MOI) of the phage. 

### 2.6. Electron Microscopy

For negative staining, specimens were placed onto grids coated with formvar film and then, after drying, treated with 0.3% aqueous solution of uranyl acetate (pH 4.0). The specimen samples were examined with a JEM-1400 (JEOL, Tokyo, Japan) transmission electron microscope at an accelerating voltage of 80 kV.

### 2.7. Phage Genome Sequencing and Annotation

Phage DNAs were isolated using the standard phenol–chloroform method after incubation of the sample in 0.5% SDS and 50 µg/mL proteinase K at 65 °C for 20 min. Fragment genome libraries were prepared using 200 ng of genomic DNA as a starting material. DNA was fragmented by ultrasound using the Bioruptor™ sonicator (Diagenode, Liege, Belgium). Fragmented DNA was used as an input for library preparation using the NEBNext Ultra DNA Library Prep Kit for Illumina (New England Biolabs, Ipswich, MA, USA) according to the manufacturer’s instructions. The library was on the Illumina MiSeq™ platform (Illumina) using paired 150 bp reads.

De novo genome assembly was accomplished using CLC Genomic Workbench 23 (QIAGEN, Aarhus, Denmark). A search for open-reading frames (ORFs) was performed using Prokka v1.13.4 [28], Glimmer v3.0.2 [29] and Prodigal v2.6.3 [30]. ORF boundaries were curated manually. Gene functions were predicted using BLAST [31], InterPro [32] and HHpred [33]. The BLAST search used the NCBI nr/nt databases and the HHpred search used PDB70_mmcif_2023-06-18, PfamA-v35, UniProt-SwissProt-viral70_3_Nov_2021 and NCBI_Conserved_Domains(CD)_v3.19 databases. The tRNA genes were searched with ARAGORN [34]. The genome of *Xanthomonas* phage PBR31 was deposited in the NCBI GenBank under accession number MT119766.

### 2.8. Genome and Proteome Analysis

Intergenomic comparisons and calculations of intergenomic similarities were performed using clinker [35] and VIRIDIC [36] with default settings. Genetic maps and gene comparisons were visualised in clinker. Protein sequences alignments were made using MAFFT [37] and the L-INS-I algorithm. Phylogenetic analysis was performed using IQ-TREE v2.2.5 [38] and “--alrt 1000 -B 5000” command line parameters. The resulting consensus trees with bootstrap support values (1000 replicas) were visualised using iTOL v6 [39]. The proteomic tree was obtained using ViPTree [40].

Protein structures were modelled with AlphaFold 2.2.4 (AF2) [41] using full databases and the command line parameters “--monomer” (for monomeric protein) and “--multimer” (for protein complexes). A search for similar structures was conducted using the DALI server [42] and best-ranked AF2 predicted structure.

The gene-sharing network was created using the vConTACT.2.0 pipeline [43] and the INfrastructure for a PHAge REference Database (INPHARED) from 1 October 2023, downloaded from https://github.com/RyanCook94/inphared/tree/main (accessed on 15 October 2023) [44]. The results were visualised using Cytoscape v3.9.0 [45].

## 3. Results

### 3.1. Bacteriophage Isolation and Host Range Determination

*Xanthomonas* phage PBR31 was isolated from a mixture of soil and stumps after harvesting cabbage in the fields of the Transnistrian Agricultural Research Institute (Tiraspol region, Moldova) in October 2014. An identical phage was isolated from soil in the Moscow region, Russia, in October 2015, using the Ram3-1 strain as a host. Phage lytic activity was tested against fourteen strains of *Xcc* from the collection of the Russian State Agrarian University—MTAA. Lytic activity was revealed for ten strains tested (Table 1) on the upper agar. The phage formed small cloudy plaques (Ø1–2 mm) with vague borders and an irregular shape (Figure 1a). The morphology of phage PBR31, as shown by transmission electron microscopy (Figure 1b), can be classified as podovirus morphotype C1 with an icosahedral head about 60 nm in diameter measured face-to-face and a short, non-contractile tail. Morphologically, PBR31 resembles podophages *Escherichia* phage T7 and *Salmonella* phage P22 [46,47].

### 3.2. Resistance to Stress Factors and Kinetic Features

The phage was characterised by a fairly long adsorption time on host cells. Thus, phage particles were attached to cells almost completely (on average 88.6%) only after 60 min of cultivation (Figure 2A). Phage PBR31 lysed cells within 180 min and produced 28.3 ± 7.5 virions per infected bacterial cell (Figure 2B).

Phage PBR31 was resistant to high concentrations of chloroform (Figure 3A). Even in 75% chloroform solution, the phage retained about 9% activity. Phage particles reduced their titre by 96.4% at a temperature of 50 °C and the complete loss of phage viability occurred at 60 °C with an exposure of 1 h (Figure 3B). The pH values of 6–9 were optimal for the phage, while a pH of 3–5 and a pH of 10–12 resulted in partial or complete loss of viability (Figure 3C). The decrease in phage titre correlated with exposure to UV treatment (Figure 3D). Thus, complete destruction of phage virions occurred after 40 min of exposure and a noticeable decrease was observed after 30 min (by 99.7%, compared with the initial titre). An experiment to determine the optimal MOI showed that, at a concentration of phage virions in the starting mixture of 0.01 PFU/CFU, the phage yield after 24 h of cultivation was maximal and amounted to 2.7 × 10^7^ PFU/mL (Figure 3E).

### 3.3. General Characterisation of the Genome

*Xanthomonas* phage PBR31 is a double-stranded DNA virus. According to genomic data, the phage virion is characterised as a Podoviral morphotype. The length of the PBR31 genome is 39,980 base pairs (bp) (GenBank accession #MT119766). The GC content of the genome is 55.3%. This number is noticeably lower than the usual GC content of *Xcc*, which is about 65%. Gene prediction tools identified 71 ORFs. No tRNA genes were found in the genome. A BLAST search combined with HHpred suggested putative functions for 52 predicted proteins, and 29 genes were annotated as encoding hypothetical proteins. Counting from the 5′-end, 40 genes are oriented in a forward direction, with the remaining genes located in the opposite strand, which is reminiscent of phage λ (the genus *Lambdavirus*) and other temperate phages. In experiments using susceptible bacterial strains, the phage demonstrated lytic behaviour, but the presence of integrase and excisionase genes suggests the possibility of integration into the host chromosome and the ability to develop temperate lifestyle. In addition, BLAST searches using MCP, TLS and integrase sequences revealed the presence of apparently related genes in prophage regions of various bacteria, including *Xanthomonas* strains. These prophage regions also contain other genes characteristic of temperate podophages, including podoviral internal and tail tube proteins.

The PBR31 genome has a modular structure. Counting from the 5′-end of the genomic assembly, the genome contains a group of genes that may be associated with lysogeny regulation and replication (Figure 4). Evidence of this is the presence of genes encoding proteins similar to repressors of temperate phages, including the well-studied phages λ and P22 (the genus *Lederbergvirus*), as well as the presence of proteins showing remote homology to phage λ replication proteins O and P. In phage λ, they are involved in the initiation and propagation of replication forks [48]. This block also contains other genes involved in DNA manipulations or performing regulatory functions. Unlike many λ-like phages, phage PBR31 does not contain a gene encoding a bifunctional DNA primase/helicase, nor does it contain genes encoding common phage DNA polymerases (DNAPs) belonging to family A (like phage T7) or family B (like phages ϕ29 and T4) [49,50,51]. Interestingly, the PBR31 genome includes gene 17 (g17) encoding a protein with a remote but clear homology to a small subunit of archaeal D-family DNAP II. DNAP II can function as a DNA polymerase and an exonuclease that degrades single-stranded DNA in the 3′ to 5′ direction [52]. In the genomes of known phages, homologue genes found with BLAST searches are mainly annotated as protein phosphatase, oxidoreductase and hypothetical proteins, but the HHpred analysis identified gp17 (gene product 17) as a small subunit of D-family DNAP II (probability 99.59%, E-value: 1.9 × 10^−13^). Gene 17 is adjacent to a gene encoding a small putative iron–sulphur protein of 70 aa (amino acids) located upstream of g17. This putative iron–sulphur protein may also be involved in phage replication. In eukaryotes, iron–sulphur clusters provide structural stability to the catalytic subunit of DNA polymerase δ, which is evolutionarily related to archaeal D-family DNA polymerases [53,54]. To date, D-family DNA polymerases have only been identified in Euryarchaeota [49].

The packaging block includes genes encoding the small and large subunits of terminase. The large subunit of terminase (TLS, terminase large subunit, terminase) is a two-domain protein containing an N-terminal ATPase domain and a C-terminal nuclease domain [55]. Interestingly, a DALI structural search using the PBR31 ATPase domain of terminase pointed to the structure of *Salmonella* phage Sf6 (*Lederbergvirus Sf6*, PDB code 4IEE) as the most similar structure (DALI score 31.0, RMSD 1.2 Å), whereas the structural search using the nuclease domain pointed to *Geobacillus* phage D6E terminase (PDB code 5OE9) as having the closest structure (DALI score 15.7, RMSD 3.3 Å). A BLAST search identified homologues of the PBR31 ATPase domain sequence among representatives of *Lederbergvirus* phages but found no such homologues for the nuclease domain sequence. This may be related to the horizontal transfer of parts of TLS genes corresponding to different domains and/or to a higher rate of evolution of the nuclease domain.

The structural block of genes is similar to that of *Autographiviridae* (T7-like), *Lederbergvirus* (P22-like) and other phages characterised by Podoviral morphology. The portal protein (PP) and major capsid protein (MCP) also show sequence similarity with proteins of other Podoviruses. As expected, AlphaFold 2 modelling showed that the PBR31 MCP adopts the HK97-like fold characteristic of tailed phages, herpesviruses and mirusviruses [56] (Figure 5a). A DALI search using the PBR31 MCP identified the structure of *Salmonella* phage Sf6 (*Lederbergvirus Sf6*, PDB code 5L35) as the closest structure (DALI score 32.9, RMSD 2.8 Å). A structural comparison showed that the structural architecture of PBR31 MCP (as well as *Lederbergvirus* MCPs) differs from phage HK97 MCP by the presence of an additional subdomain in the region of the E-loop and an additional α-helix in the A-domain (Figure 5a). It is noteworthy that, despite the high degree of structural similarity of the major capsid proteins of phages PBR31 and Sf6, their pairwise sequence identity is only about 25%. The structural block also includes genes encoding the tailspike protein (TSP, tailspike) (Figure 5b), which apparently acts as a receptor-binding protein (RBP). AF2 modelling identified the structural architecture of the PBR31 tailspike as being three-domain. The DALI search revealed similarities between the PBR31 TSP, various cell wall-degrading enzymes and tailspikes of viruses infecting *Salmonella* and other enterobacteria, including *Lederbergviruses*. Resembling the TSPs of *Salmonella* phages Sf6 and P22 (*Lederbergvirus P22*), the N-terminal part is composed of β-sheets and apparently is the virion-binding domain [57,58]. The central domain has a triple-β-helix fold and is possibly responsible for binding with the receptor. HHpred found similarities between this central domain and pectate lyase superfamily proteins (Pfam code PF12708). The RBPs of phage P22 TSP also have pectate lyase-like central domains. The C-terminal part of PBR31 TSP has a β-prism structure and can be important for oligomeric assembly of the TSP trimer, as in phage P22 [59].

The lysis module of phage PBR31 encodes a lysis machinery, which can be more advanced than the usual three-step lysis system [61]. The genome lysis block includes adjacent genes that have been reliably identified as encoding holin, endolysin (Lys) and spanin (gene 38). In addition, the PBR31 genome contains gene 39, encoding an unidentified protein 75 aa gp39, and gene 40, encoding a 215 aa esterase belonging to the SGNH-hydrolase family. AF2 modelling predicted the structural architecture of gp39 to consist of three α-helices connected by linkers. Hypothetically, gp39 could be a periplasmic protein that functions as o-spanin together with gp38, which acts as i-spanin [62]. In turn, the esterase gp40 contains an N-terminal signal peptide and shows a remarkable level of similarity to phage tail-associated hydrolases (HHpred probability up to 99.75%) and cellular lipolytic enzymes (HHpred probability up to 99.86%). Perhaps gp40 facilitates lysis by using its hydrolase cell wall-degrading activity.

### 3.4. Phylogenetic and Phylogenomic Analyses

Phylogenetic analysis was performed using the sequences of major capsid proteins, portal proteins, large subunits of terminase and endolysins. TLSs, PPs and MCPs are among the most conserved phage proteins [63,64,65]. Endolysin was taken to represent the lysis module. To construct the trees, 120 representative genomes were taken from the results of BLAST searches and the GenBank phage database, and genes encoding MCP, PP, TLS and EL were predicted and verified using HHpred. Interestingly, calculations of intergenomic nucleotide similarity did not show a meaningful level of similarity between phage PBR31 and representative genomic sequences (Appendix A). 

The trees constructed have different topologies (Figure 6). This may be a consequence of horizontal exchanges of genes and genetic blocks, and illustrates the modular evolution of bacteriophages [66]. However, the composition of branches of MCP and PP trees, containing phage PBR31, is similar, and the topologies of these trees are similar, which may indicate the simultaneous transfer of genes encoding the structural proteins of capsid. In these trees, phage PBR31 is close to uncharacterised phage *Podoviridae* sp. isolate ctn9Y15, temperate Podophages *Acetobacter* phage ϕAX1 [67], *Pseudomonas* phage PAE2 [68], lytic Podophage *Rhodoferax* phage P26218 [69], temperate Podophage Xfas53 infecting xylem-inhabiting bacterium *Xylella fastidiosa* [70] and, presumably, temperate Podophage *Vibrio* phage VvAW1 [71] (these are listed in order of increasing distance from PBR31 in the MCP tree). These phages infect evolutionarily distant bacteria inhabiting different ecological niches. With high bootstrap support, the MCP and PP trees place the branches containing the phages listed above in a large clade that also contains temperate Podophages *Shigella* phage Sf6 (*Lederbergvirus Sf6*) [72] and *Ralstonia* phage RSK1 (*Firingavirus RSK1*) [73]. It is noteworthy that the PP tree shows better bootstrap support than the MCP tree, which is possibly related to the higher degree of conservation of the portal protein compared with the major capsid protein.

The phylogenetic trees constructed using sequences of terminase and endolysin suggest the different evolutionary history of these two proteins. In the TLS tree, the phage PBR31 is placed in the same branch as the uncharacterised Myovirus *Rhizobium* phage RHph_N17 that presumably belongs to the genus *Kleczkowskavirus*. The endolysin tree groups phage PBR31 together with Siphovirus *Psychrobacter* phage Psymv2 [74]. Both TLS and Lys trees place *Podoviridae* sp. isolate ctn9Y15 and *Shigella* phage Sf6 distantly from phage PBR31. Interestingly, in these trees, phages ctn9Y15 and Sf6 are located not far from each other.

Phylogenomic analysis was conducted using the ViPTree server. ViPtree uses tBLASTx searches to find homology between the query genome and the genomes included in the ViPTree database. Using genomic sequences and tBLASTx algorithms, the resulting tree can be called a “proteomic tree”. It was shown that viral groups identified in a proteomic tree correspond well to official classifications [40]. Because the ViPtree uses its own database, it is difficult to compare topologies of single-protein trees and the ViPtree proteomic tree. The ViPtree proteomic tree (Appendix A) does, however, show similarities to single-protein trees in clade composition, placing phage PBR31 in the same clade as *Rhodoferax* phage P26218 and other phages neighbouring PBR31 in the MCP and PP trees. In addition, the ViPtree proteomic tree indicates the relatedness of phage PBR31 and *Lederbergvirus* phages.

### 3.5. Gene Network Analysis and Intergenomic Comparisons

Evolutionary relations between *Xanthomonas* phage PBR31 and known phages with sequenced genomes contained in the INPHARED database were also studied using a network-based approach implemented in vConTACT v.2.0. Nearly identical (96%) replication of existing genus-level viral taxonomy assignments from ICTV has been reported using this pipeline. The analysis used the genomic sequences of phage PBR31 and 120 representative genomes, which showed the clustering of a group of 33 phages, including PBR31 (PBR31 cluster), on a separate island (Figure 7a). There are 28 phages that are represented by complete genomic sequences and 5 phages (EBPR Podovirus 1, *Vibrio* phages 1.183.O._10N.286.48.B7, 1.184.A._10N.286.49.A5, 1.211.A._10N.222.52.F11 and 1.211.B._10N.222.52.F11) are represented by nearly complete genomes. All phages are unclassified. Most members of the PBR31 cluster are bacteriophages characterised by a temperate lifestyle. The genomic size of all sequences varies from 33,272 to 53,192 bp. According to the genomic data, all phages, except *Komagataeibacter* phage ϕKX1, are characterised by Podoviral morphology; *Komagataeibacter* phage ϕKX1 apparently has a Myoviral morphology. According to NCBI records, two phages in the PBR31 cluster are designated *Myoviridae* spp., but the genomic analysis identified them as Podoviruses.

Notwithstanding the results of vConTACT clustering, calculations of intergenomic nucleotide similarity using the VIRIDIC pipeline did not show a meaningful level of similarity between phage PBR31 and other members of the vConTACT PBR31 cluster (Figure 7b). VIRIDIC is an intergenomic distance calculator designed to group viruses using an algorithm that implements the clustering method traditionally used by ICTV [36]. ICTV has established the 70% nucleotide identity of the full genome length as the ‘cut-off’ for genera. Seemingly, classification of PBR31 based on the ICTV rules (“taxonomic classification”) requires the creation of a new taxon at a rank of genus or higher.

To verify the assumption about the mosaic architecture of the PBR31 genome, an alignment of PBR of genomes of PBR31 and other phages was carried out (Figure 8). Protein sequences of the nearest neighbours of phage PBR31, identified using single-sequence phylogeny, BLAST searches and ViPtree phylogeny, were taken for alignments. Genome alignments showed that homologous regions of the genomes analysed can include only one gene (as in the case of *Aminobacter* phage Erebus containing the closest homologue of PBR31 terminase) or several genes (as in the case of structural genes of phage PBR31, *Podoviridae* sp. isolate ctn9Y15, *Pseudomonas* phage PAE2 and *Vibrio* phage vB_ValP_FGH). In some cases, the genomic architecture has a pronounced chimeric structure, where about a third of the phage genome is similar to the genomic regions of other phages, while the other part(s) are “acquired” from an ancestor from another lineage (*Pseudomonas* phage HU1 and AF, *Shewanella* phage X14 and *Vibrio* phage VvAW1). Interestingly, the composition of a gene block transferred during recombination events that leads to the creation of a mosaic genome does not necessarily include only functionally related genes. For example, the alignment of PBR31 and phage *Podoviridae* sp. isolate ctn9Y15 shows a similarity between the genes of capsid proteins and the portal proteins, but shows no homology between the TLS genes located upstream of the PP, although TLS generally seems to be more conserved than MCP [75]. At the same time, homologous gene blocks of *Podoviridae* sp. isolate ctn9Y15 and *Pseudomonas* phage PAE2 include both structural genes and terminase (Figure 8, upper scheme). The PPs, MCPs and TLSs of phages ctn9Y15 and PAE2 have a similar level of pairwise identity of about 37–38%.

## 4. Discussion

*Xcc* is an economically significant pathogen in crop production, causing reductions in yield and product quality in brassica crops. To date, some bacteriophages active against *Xcc* have been described [16,76], but characterised lytic phages outnumber temperate ones. The increased interest in lytic phages is understandable, since they can be used for the biological control of disease in crop production. In this regard, information about temperate phages of *Xcc* is insufficient, although temperate phages, according to precedents in other species, can play an important role in promoting the diversity of the host bacterium and the emergence of new strains through horizontal gene transfer [77,78,79].

Compared with genetically related phages, PRB31 has been shown to be phenotypically close to the *Rhodoferax* phage P26218 (based on the same plaque size) [69] and differs to a lesser extent from the *Xylella* phage Xfas53 in the rate of adsorption on host cells [70]. According to kinetic characteristics, the virus is close to the *Vibrio* phage VvAW1, with a similar adsorption time (about 70 min), and has a weak effect on the growth rate of the bacterium due to its moderate nature [80]. From a more genetically distant phage, for example, from the *Salmonella* phage BTP1 (the genus *Lederbergvirus*), PRB31 was isolated due to its much smaller burst size [81].

Genomic analysis identified phage PBR31 as a temperate bacteriophage. The PBR31 genome contains the genes apparently related to lysogenic interactions, including the genes encoding integrase, excisionase and repressors. The PBR31 proteins related to a temperate lifestyle do not exhibit a strong homology with those of the well-studied λ-like phages or the temperate P22-like Podophages assigned to the genus *Lederbergvirus*, but they are distantly related to them based on the results of remote homology detection. This is reminiscent of the observations made in studies of λ-like MD8-like phages infecting *Pseudomonas* [15]. The genome architecture of PBR31 is similar to that of *Lederbergvirus* phages. The relatedness of phage PBR31 and *Lederbergvirus* phages was also supported by phylogenetic analysis using sequences of major capsid and portal proteins, as well as the analysis of predicted structures of a major capsid protein and a large subunit of terminase. Taxonomic classification of PBR31 does not, however, appear to be straightforward.

The ICTV Subcommittee for Bacteriophage and Archaeal Viruses proposed a ‘cut-off’ of 70% nucleotide identity as a criterion for distinguishing phage genera. According to this criterion, phage PBR31 should be unambiguously assigned to a new genus, although it appears to be very difficult to propose a consistent classification scheme that includes higher-ranking taxa at the level of subfamilies and families. According to guidelines for the demarcation of species-, genus-, subfamily- and family-level ranks of tailed phage taxonomy, subfamilies should “share a low degree of nucleotide sequence similarity and that the genera form a clade in a marker tree phylogeny” and “the family is represented by a cohesive and monophyletic group in the main predicted proteome-based clustering tools (ViPTree, GRAViTy dendrogram, vConTACT2 network)” [7]. The topologies of phylogenetic genes can, however, be different for such important signature genes as the MCP of HK97-fold and TLS, which are hallmarks of the class *Caudoviricetes*. In addition, multidomain proteins may have domains acquired by horizontal transfer, as may have occurred in the case of the PBR31 terminase domains, and which is pronounced for phage RBPs [82]. It is also difficult to accept proteome-based phylogenies when individual sequence phylogenies are inconsistent. In addition, it is easy to imagine a hypothetical situation where a recent gene transfer into the viral genome could radically change the topology of the tree based on weak signals caused by the early divergence of a unique group.

Another issue concerns the consistency of results among different proteome-based clustering tools. In the case of PBR31, the ViP tree groups PBR31 and *Lederbergvirus* phages into one clade, but the vConTACT network places PBR31 in a distinct cluster that does not contain a representative of the genus *Lederbergvirus*. The authors speculate that this could be a consequence of the influence of gene exchange, due to which it is unlikely to be possible to construct a classification scheme based on sequence similarity where there are low levels of this similarity.

The classification of tailed phages at the level of families and subfamilies has two fundamental problems, one related to the rapid divergence of phage proteins and the other to the mosaic nature of phage genomes. Apparently, only the former problem can be at least partially solved by an analysis of protein folding and the similarity of experimentally determined or predicted structures of phage proteins [75,83]. A solution to the problem of the classification of phages with a pronounced mosaic genome requires re-estimation of the principles of classification and reconsideration of the applicability of bioinformatic tools used in classification. Previous work [15] has suggested using genome architecture as the key feature to create phage families with chimeric genomes, but this approach also requires the development of rules for assessing and classifying genomic architecture. 

## 5. Conclusions

*Xanthomonas* PRB31 is a temperate bacteriophage that shows no significant genomic sequence similarity to known phages. Based on the results of intergenomic nucleotide comparison, phage PBR31 should be defined as a representative of a new genus. Phage PBR31 shows similarities in genomic architecture, lifestyle and morphology with *Lederbergvirus* phages. Taxonomic classification of PBR31 and related phages at the level of subfamilies and families is, however, extremely difficult due to the mosaic nature of the genomes of these phages. High mutation rates and extensive lateral transfer raise questions about the feasibility of classification algorithms for temperate phages, most of which either remain unclassified or have been classified only at the genus level. The genome of PRB31 encodes tailspikes and lysins possessing cell wall-degrading activity, which can be used for the treatment of infection caused by *Xcc*.

## Figures and Tables

**Figure 1 viruses-16-00406-f001:**
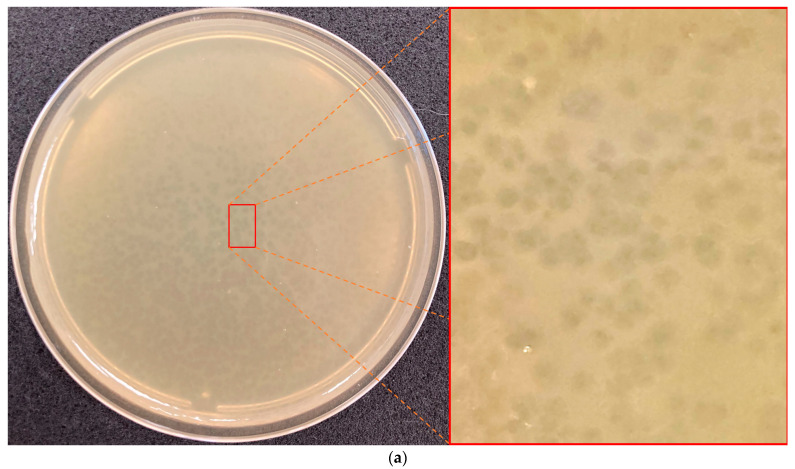
(**a**) Plaque morphology of the phage PBR31 on King’s B top agar with *X. campestris* pv. *campestris* Ram3-1 strain. (**b**) Transmission electron microscopy of bacteriophage PBR31. Staining with 1% uranyl acetate. The scale bar is 100 nm.

**Figure 2 viruses-16-00406-f002:**
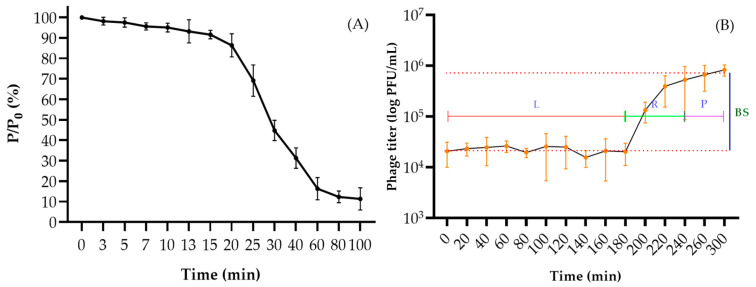
Phage adsorption curve (**A**) and one-step phage growth curve (**B**) of *X. campestris* pv. *campestris* phage PBR31. Strain Ram3-1 was used as a host. The y-axis shows the ratio of the current titre, at each time point (P), to the initial value (Po), multiplied by 100%. The y-axis shows the ratio of the current titre, at each time point (P), to the initial value (Po), multiplied by 100%. L—latent phase; R—virion release phase; P—plateau phase; BS—burst size. Values in panels represent the mean of three independent trials and error bars represent the standard deviation.

**Figure 3 viruses-16-00406-f003:**
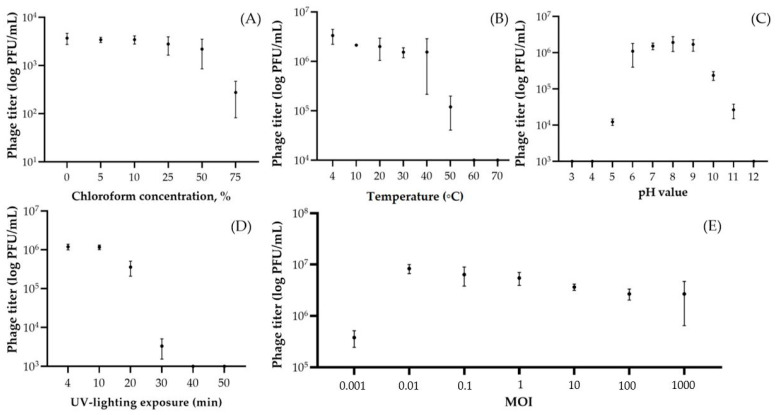
Survival of *X. campestris* pv. *campestris* phage PBR31 under various stress factors and optimal multiplicity of infection. Phage solutions were mixed with chloroform at a concentration of 5–75% (**A**) and treated with temperature increases from 4 to 100 °C for 1 h (**B**), with the pH varying from 3 to 12 for 1 h (**C**) and with ultraviolet irradiation for 4–50 min (**D**). Comparison of phage titre after 24 h incubation at seven MOI ratios (0.001, 0.01, 0.1, 1, 10, 100 and 1000 PFU/CFU) (**E**). All tests were repeated three times. Standard deviation (sd) is shown for each bar.

**Figure 4 viruses-16-00406-f004:**
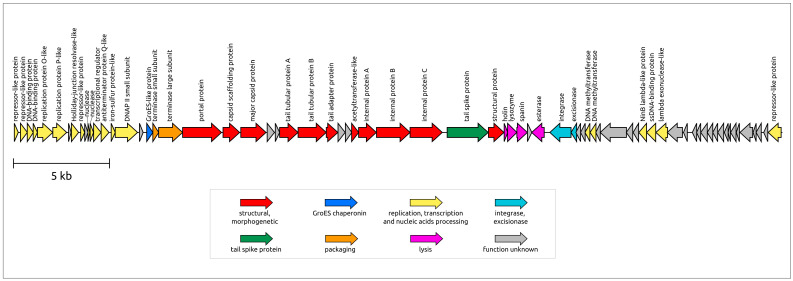
Genetic map of *Xanthomonas* phage PBR31. Arrows indicate the direction of transcription. The scalebar indicates the length of the nucleotide sequence. Gene functions are shown in labels and the legend.

**Figure 5 viruses-16-00406-f005:**
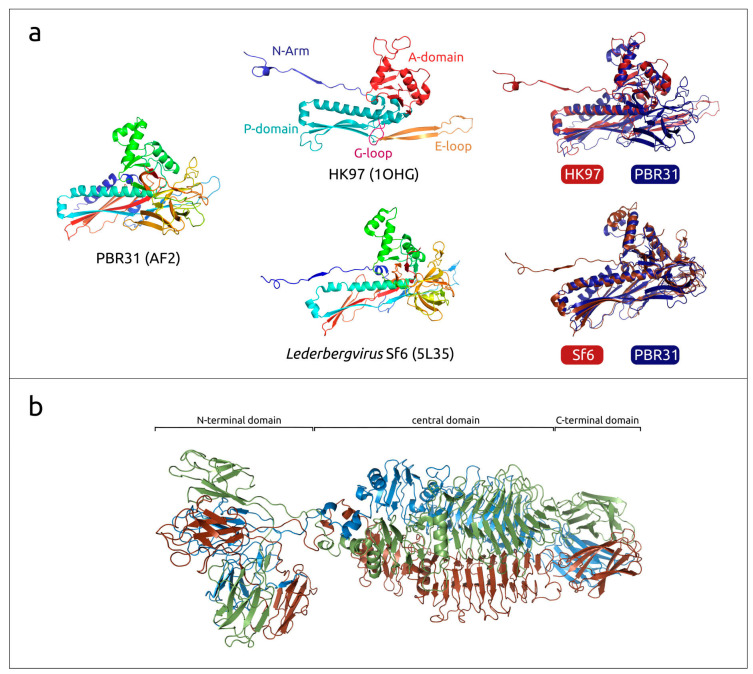
(**a**) Ribbon diagram of the predicted PBR31 MCP and experimentally determined structure of phage Sf6 MCP (PDB code 5L35), coloured based on a rainbow gradient scheme, where the N-terminus of the polypeptide chain is coloured blue and the C-terminus is coloured red; the HK97 MCP (PDB code 1OHG) and its common features are coloured as indicated in the figure [60]; the superimposition of HK97 and Sf6 MCPs to PBR31 MCP is coloured as indicated in the figure. (**b**) Ribbon diagram of the predicted PBR31 TSP trimer, where each monomer is identified using a different colour.

**Figure 6 viruses-16-00406-f006:**
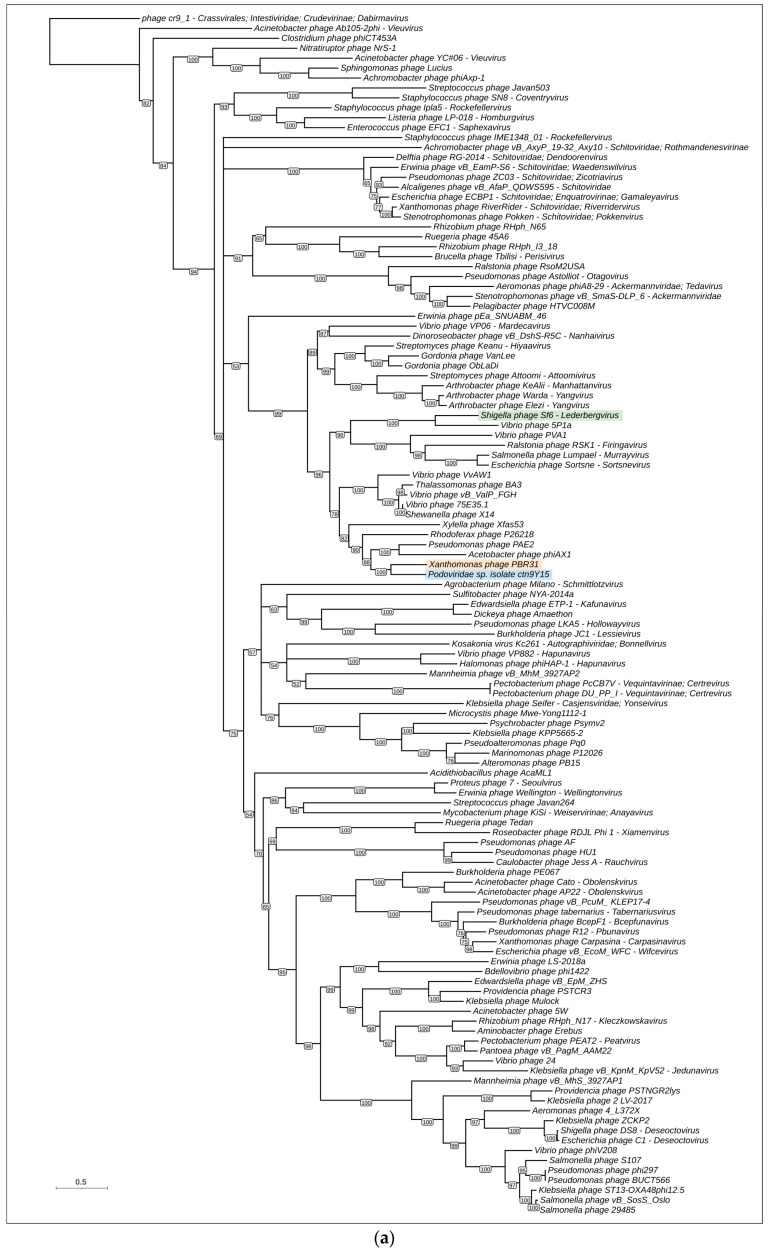
Maximum likelihood phylogenetic trees based on amino acid sequences of MCP (**a**), PP (**b**), TLS (**c**) and Lys (**d**). *Xanthomonas* phage PBR31 is highlighted in orange, phage *Podoviridae* sp. isolate ctn9Y15 is highlighted in blue and *Salmonella* phage Sf6 is highlighted in green. Bootstrap values are shown near their branches. The scale bar shows 0.5 estimated substitutions per site and the trees were rooted to phage cr9_1.

**Figure 7 viruses-16-00406-f007:**
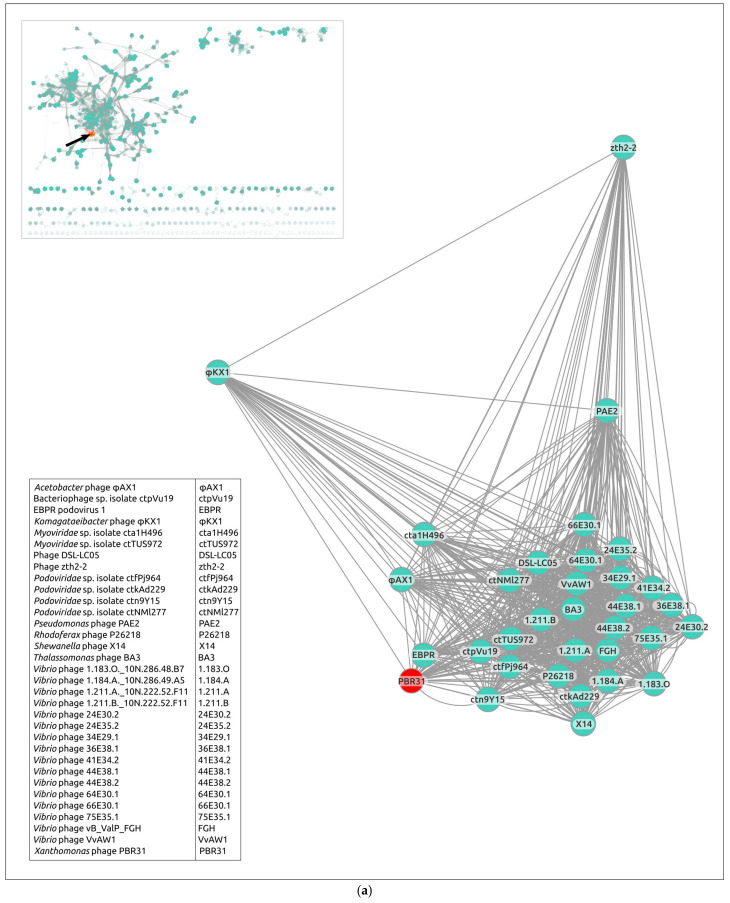
(**a**) Gene-sharing network cluster containing *Xanthomonas* phage PBR31 and other phages, created using the vConTACT 2.0 and INPHARED database. The general view of the gene-sharing network is shown in the upper-right corner, with the PBR31-cluster coloured red. (**b**) VIRIDIC-generated heatmap of *Xanthomonas* phage PBR31 and phages belonging to the vConTACT PBR31 cluster. The colour coding in the upper-right part of the map indicates the clustering of the phage genomes based on intergenomic similarity. Numbers represent similarity values for each genome pair, rounded to the first decimal. The aligned genome fraction and genome length ratio are shown in the lower-left of the map, using the colour gradient that is explained in the legend.

**Figure 8 viruses-16-00406-f008:**
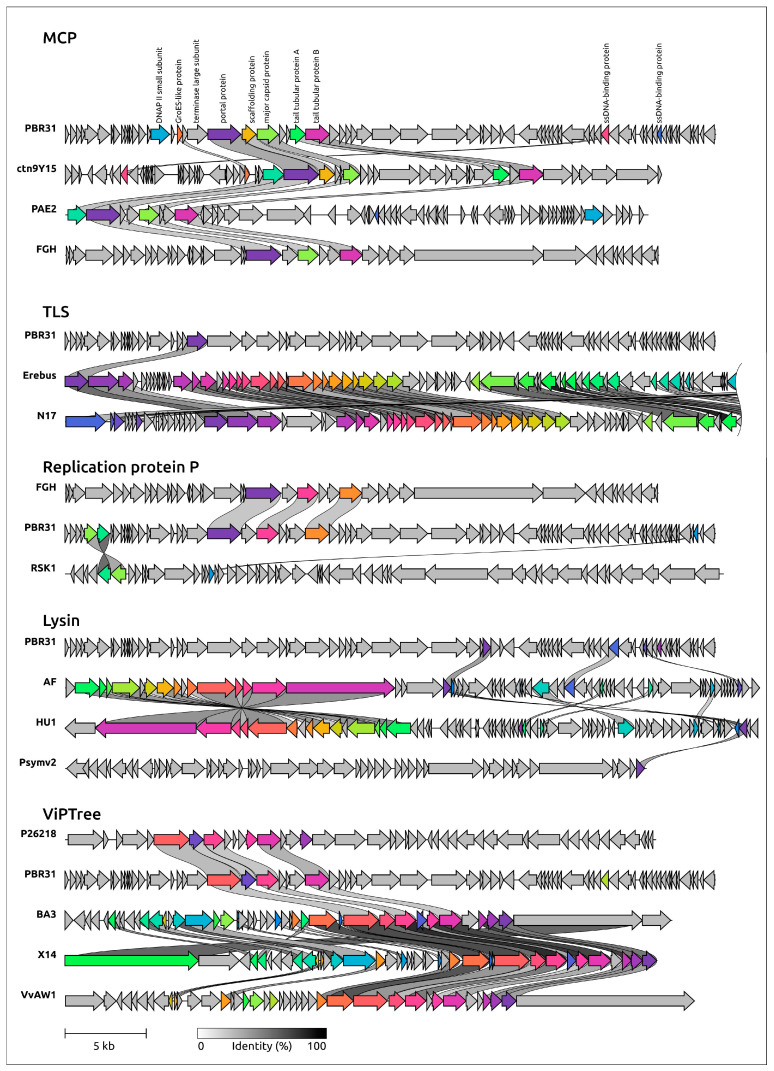
Comparative genome alignment of *Xanthomonas* phage PBR31 and phages *Podoviridae* sp. isolate ctn9Y15 (ctn9Y15), *Pseudomonas* phage PAE2 (PAE2), *Vibrio* phage vB_ValP_FGH (FGH), *Aminobacter* phage Erebus (Erebus), *Rhizobium* phage RHph_N17 (N17), *Ralstonia* phage RSK1 (RSK1), *Pseudomonas* phage AF (AF), *Pseudomonas* phage HU1 (HU1), *Psychrobacter* phage Psymv2 (Psymv2) and *Rhodoferax* phage P26218 (P26218). Percentage of amino acid identity is represented by greyscale links between genomes. Homologous proteins are assigned a unique colour.

**Table 1 viruses-16-00406-t001:** The spectrum of lytic activity of the phage PBR31 against *X. campestris* pv. *campestris* strains.

Name of the Strain	Date of Isolation	Place of Origin, Plant	Genbank 16S №	Lysis Zone Caused by Phage R3-1
BK-55	10.2017	Krasnodar region, Russia, white cabbage	OR626094	+
CK-71	10.2017	Krasnodar region, Russia, cauliflower	OR626097	+
Xcc 1/1	09.2017	Moscow region, Dmitrov, Russia, white cabbage	OR626648	+
Bes-1	09.2016	Moscow region, Dmitrov, Russia, white cabbage	OR626092	-
Cas	09.2016	Moscow region, Dmitrov, Russia, cauliflower	OR626095	+
Tr1	11.2012	Tiraspol, Transnistria, Moldova, cabbage	OR626099	+
DK-1	10.2012	Moscow region, Serpukhov, Russia, white cabbage	OR626096	+
Ram 3-1	10.2012	Moscow region, Ramensky, Russia, cabbage	OR625211	+
XУ 1-2	10.2012	Ukraine, white cabbage	OR644606	-
Bel-2	10.2006	Belarus, white cabbage	OR626091	-
Bun-1	09.2006	Moscow region, Dmitrov, Russia, white cabbage	OR626093	+
Xn-13	1997	Japan, Mie-ken, *Capsélla búrsa-pastóris* (shepherd’s purse)	OR626098	+
306NZ	-	The Netherlands	OR626090	+
NCPPB 528T	1957	UK, cabbage	-	-

## Data Availability

Data are contained within the article and Appendix A.

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
