# Peer review of "Xanthomonas Phage PBR31: Classifying the Unclassifiable"

_viruses, 2024, doi:10.3390/v16030406_

Round 1
Reviewer 1 Report
Comments and Suggestions for Authors
Although this is an interesting study describing a novel phage of an important crop pathogen, I felt the manuscript could do a better job of presenting this work - both cleaning up and shortening some sections and elaborating in a few places. Almost all figure captions need more details (if they are supposed to be read independent of text)
In particular the problem of classifying phages is not unique to this phage and the fact that this phage cannot be easily classified is not particularly surprising. I would shorten a lot of that description (e.g., lines 60-87 in introduction, much of discussion). I'm not sure what figures 7 and 8 add to manuscript. Figure 9 is impossible to interpret.
TEM is mentioned in methods but no TEM picture provided. Figure 4 and 9 are hard to read.
I'm not a big fan of the phylogenetic trees that cannot be read. In addition, I could not figure out what model was used (Parsimony, maximum likelihood, Bayesian?). I would collapse all the branches with less than 50% support. I would better justify chose of genes. I would also only present a much reduced/close-up of each tree with branches relevant
A lot is made that this phage is temperate - a conclusion supported by its annotation (and it's excise not excisions since this is not an enzyme). I was thus surprised to not see any lysogen test/assay. I would have found that more interesting than the data on phage stability under different conditions; in particular if the test included all the strains in Table 1.
In summary, it's obvious that the authors have done a lot of work and there are some interesting stuff in this manuscript but the authors have not figured out what story they want to tell about this phage.
Comments on the Quality of English LanguageOverall the English is good, although there a numerous places where the language could be cleaned up. It's hard to give more specific advice since my opinion is that this manuscript needs to be re-organized.
Author Response
Dear Reviewer 1, thank you for your positive feedback on our manuscript. We appreciate your efforts to improve the manuscript. Please find below a response.
- The Introduction has been shortened.
- TEM image has been provided (Figure 1b).
- Figure 7 was moved to Supplementary materials. Figures 7 and 8 show phage clustering, they are often required for classification and VIRIDIC clustering is present in almost all recent papers on phage genomics and classification.
- All branches with less than 50% support have been collapsed. “Maximum likelihood” was added to tree captions.
- These genes are shared by Caudoviricetes viruses which can enter the infection cycle without the help of other phages. TLSs, PPs and MCPs are among the most conserved phage proteins. Endolysin is taken to represent the lysis module. We have added this note and corresponding references. (Lines 411-413).
- In our experiment, the phage PBR31 showed lytic behavior towards several sensitive strains of Xanthomonas campestris. The genomic data suggests an ability of PBR31 to temperate lifestyle. Besides, using the sequences of MCP, TLS and integrase we found related prophages in different bacterial genomes including Xanthomonas Corresponding prophage regions contain the gene characteristic for temperate podoviruses including integrase and podoviral internal and tail tube proteins. We added this information to the text (Lines 313-319). We plan on to continue the study of PBR31 and discuss the results of the analysis of other aspects of interaction in the phage-bacterium host system in the next work.
- “it's excise not excisions since this is not an enzyme” – Corrected, thank you (Lines 562).
- We intended to characterize the phage which represented a distinct group of podophages distantly resembling P22-like phages (Lederbergviruses) and to underline the problems of classification of this phage. We tried to show that these problems are mainly related to genetic mosaicism and hopefully could do it. We think this issue is interesting for researchers interested in phage taxonomy and evolution.

Reviewer 2 Report
Comments and Suggestions for Authors
This is a very thorough and well written manuscript. However, the pdf file consists of 32 pages. The paper could be shortened by deleting Figures 6 and 7 since the negative results are adequately described in the text and corroborated by Figure 8.
One typo was observed: Line368 PBR31 should be not RPB31
Author Response
Dear Reviewer 2,
Thank you for your high estimation of our manuscript. We moved Figure 7 to Supplementary materials. Bu if you do not mind, we would like to keep Figure 6 in the main part of manuscript since it shows evolutionary history of different phage proteins, which is important for discussion and conclusions. Besides, without both Figure 6 and Figure 7, the manuscript will lack any phylogenies.
The typo has been corrected, thank you.

Reviewer 3 Report
Comments and Suggestions for Authors
Bacteriophages are difficult to taxonomically describe and genomically analyze due to their unusual evolution. In this study, Tarakanov et al. identified a temperate bacteriophage called PBR31 from Xanthomonas through extensive genomic analysis. They found that PBR31 shares similarities with Lederbergvirus phages, but its taxonomic classification is challenging at the subfamily level. Additionally, their bioinformatics analysis revealed that PBR31 has cell wall degrading enzymes, which could have potential applications in treating bacterial infections. Overall, the manuscript is well-written and the data could support the conclusion. However, two major issues require attention.
Major issues:
Line 196. Please note that the electron microscopy data for the PBR31 phage is missing. Please provide the negative staining image of the PBR31 phage and a detailed table that shows the approximate dimensions of the capsid length, capsid width, tail length, and tail thickness of the PBR31 phages based on the electron microscope image. Additionally, please include a comparison of the PBR31 phages with other phages based on physical analysis.
Line 282. Please provide a comprehensive table listing the genomic features of the PBR31 phages. (NCBI accession number, genome size, GC content, protein prediction…)
Author Response
Dear Reviewer 3,
Thank you for your high estimation of our manuscript. TEM image has been provided. Unfortunately, we do not have sufficient data to calculate the exact tail dimensions, but general morphology is clearly podoviral, the capsid is isometric and about 60 nm in diameter which is close to podophages T7 and P22. We added these comparisons.
The table listing the genomic features of phage PBR31 provided (Supplementary Table 1).

Round 2
Reviewer 3 Report
Comments and Suggestions for Authors
The authors have addressed my concerns and made improvements to the manuscript. I support the publication of this manuscript.